# Red Ginseng Improves D-galactose-Induced Premature Ovarian Failure in Mice Based on Network Pharmacology

**DOI:** 10.3390/ijms24098210

**Published:** 2023-05-04

**Authors:** Zijing Shang, Meiling Fan, Jingtian Zhang, Zi Wang, Shuang Jiang, Wei Li

**Affiliations:** 1College of Chinese Medicinal Materials, National & Local Joint Engineering Research Center for Ginseng Breeding and Development, Jilin Agricultural University, Changchun 130118, China; szj0703@126.com (Z.S.);; 2College of Animal Medicine, Jilin Agricultural University, Changchun 130118, China; 3College of Life Sciences, Jilin Agricultural University, Changchun 130118, China

**Keywords:** premature ovarian failure, D-galactose, red ginseng, oxidative stress, apoptosis, network pharmacology

## Abstract

In this study, we evaluated the ameliorative effect and molecular mechanism of red ginseng (*Panax ginseng* C.A. Meyer) extract (RGE) on D-galactose (D-gal)-induced premature ovarian failure (POF) using network pharmacology analysis. Ginsenosides are important active ingredients in ginseng, which also contains some sugar and amino acid derivatives. We aimed to determine the key proteins through which RGE regulates POF. In this work, we retrieved and screened for active ingredients in ginseng and the corresponding POF disease targets in multiple databases. A PPI network of genes was constructed in the STRING database and core targets were screened using topological analysis. Gene ontology and Kyoto Encyclopedia of Genes and Genomes enrichment analyses were conducted in R software. Finally, molecular docking was conducted to validate the results. Female ICR mice were used to establish a POF mouse model for in vivo experiments. Serum levels of relevant estrogens were determined using ELISA and expression levels of relevant proteins in ovarian tissues were detected using immunofluorescence and western blot analysis. Network pharmacology analysis predicted that PI3K, Akt, Bax, Bcl-2, p16, and other proteins were highly correlated with POF and RGE. The results clearly showed that RGE could increase estradiol (E2) and lower follicle-stimulating hormone (FSH) levels in D-gal-fed mice. RGE restored the expression levels of related proteins by reducing Nrf2-mediated oxidative stress, PI3K/Akt-mediated apoptosis, and senescence signaling pathways. Overall, RGE has the potential to prevent and treat POF and is likely to be a promising natural protector of the ovaries.

## 1. Introduction

The ovary is the primary reproductive endocrine organ in females and plays an important role in the reproductive process and sexual behavior of mammals [1]. POF is defined by high gonadotrophin hypoestrogenic and non-physiological amenorrhea in women at least 4 months before the age of 40 years [2]. Infertility is one of the most serious clinical manifestations of premature ovarian failure [3]. It is also manifested by menopausal symptoms such as insomnia, sweating, hot flashes, tension, and anxiety. More serious manifestations include cardiovascular disease, osteoporosis, and dementia [4,5]. The causes of POF are complex and varied, and include endogenous genetic influences, autoimmune abnormalities, psychosocial factors, environmental toxins, metabolic abnormalities (such as galactose metabolism) [6], drug impairment, radiation therapy, and surgical treatment (such as oophorectomy) [7]. At present, the pathogenesis of POF is unclear [8], thus there is no effective etiological treatment. Hormone replacement therapy (HRT) is the most common treatment for POF-related symptoms [9]. However, this treatment has many side effects and increases a woman’s risk of breast and endometrial cancers [10]. Although HRT is very effective at improving the symptoms of POF, associated symptoms may return once treatment is stopped, possibly due to the imbalance in inflammation that accompanies the aging process [11].

Reactive oxygen species produced during cellular metabolism can cause cellular damage and thus lead to aging of the organism [12]. It is therefore important to maintain antioxidants and free radical scavengers in a balanced state to slow down aging of the organism. High levels of ROS and AGEs accumulate in the ovaries after D-gal administration [13], leading to impairment of the basic function of the ovary. Red ginseng is obtained by steaming fresh ginseng root (*Panax ginseng* C.A Meyer) [14], which has many pharmacological activities including immune enhancement and antitumor, antioxidant, anti-aging, anti-fatigue, and anti-diabetic activities, as well as anti-liver and anti-kidney toxicity activities [15,16]. Studies such as [17] have reported that ginsenosides Rg1 may improve fertility and reduce ovarian histopathological damage in mice by enhancing anti-inflammatory and antioxidant capacities and reducing the expression of proteins associated with the aging signaling pathway.

Network pharmacology is a method of constructing drug–target pathway networks to study drug mechanisms by mapping drug targets to specific biological nodes [18]. Network pharmacology is widely used in Chinese medicine research, including target discovery, screening for active ingredients, evaluating toxicity, mechanism research, and quality control research [19]. Network pharmacology analysis can be used to determine the effects and toxicity of drugs on specific targets and core networks, providing theoretical support for the transformation of TCM from empirical to evidence-based medicine. This study applied the network pharmacology method to analyze the potential mechanism through which RGE improves POF. In vivo experiments were then conducted to verify the findings. The use of Chinese medicine in the treatment of diseases involves multiple components, targets, and pathways, and the complexity and diversity of mechanisms likewise increases the difficulty of exploration. Therefore, the use of network pharmacology to explore the mechanisms of action of herbal medicines in the treatment of POF is appropriate, ingenious, and suitable. Several studies have shown that network pharmacology and molecular docking can successfully predict the components, targets, signaling pathways, and mechanisms of action of herbal medicines in the treatment of POF and have therefore provided theoretical basis for further exploration of the active ingredients and mechanisms of action of herbal medicines.

We hypothesized that network pharmacology analysis could identify the targets, pathways, and biological functions of RGE that are associated with improved POF. Using in vivo experiments, we verified the potential mechanism of action of RGE involved in improving POF. This study provides a basis for the development of drugs for the treatment of POF, as well as a new direction for wider application of RGE in clinical treatment.

## 2. Results

### 2.1. Visualization of Interacting Targets in Active Components of RGE and POF

We correlated drug targets with POF and obtained 21 active ingredients(Table 1) in red ginseng and 101 crossover genes (Figure 1a). These potential targets were entered into the STRING database to obtain PPI network information, and the data was subsequently entered into Cytoscape for visualization. The PPI network contains 75 nodes and 203 edges (Figure 1b). The size and color of the nodes in the PPI network represent the importance of their roles, with larger and bluer nodes representing greater relevance of the target. We used Cytoscape’s plugin CytoNCA to analyze the topology of the PPI networks. DC ≥ 4, BC ≥ 41.66, CC ≥ 0.32, EC ≥ 0.042, LAC ≥ 1.33, and NC ≥ 2.21 core targets were selected. These core targets were entered into the STRING database to capture and visualize core PPI network information (Figure 1c).

GO enrichment analysis was used to determine the biological functions of RGE associated with POF resistance in terms of biological process (BP), cellular component (CC), and molecular function (MF) (Figure 1d). The results suggested that RGE is possibly associated with activation of signaling receptors, ligands, and cytokines, and thus participates in regulating the response of ovarian cells to drugs and steroid hormones, regulating estrogen levels, and improving POF. KEGG enrichment analysis was used to explore the potential mechanism through which RGE improves POF, and the results showed that cell apoptosis, senescence, ovarian steroid generation, and other signaling pathways were highly enriched. RGE regulated biological processes through these signaling pathways and played a role in alleviating POF (Figure 1e).

### 2.2. Effects of RGE on Estrous Cycle, Estrogen, and Ovarian Histopathology

To verify the improved effect of RGE on POF, a mouse model of premature ovarian failure induced by D-gal was used. Compared with the normal control group, D-gal exposure (model group) inhibited weight gain in mice and the body weights of the mice increased steadily after treatment with RGE (Figure 2a). Changes in the estrous cycle of mice in each group were evaluated through H&E staining of vaginal smears. After two weeks of D-gal injections, the estrous cycle of mice in the D-gal group and the high-dose RGE group was disturbed and the interestrous period was prolonged. After treatment with RGE for 4 weeks, the estrous cycle of mice in the high-dose group returned to normal. Mice in the normal control group exhibited a normal estrous cycle throughout the experiment (Figure 2b,c).

Next, we investigated whether hormone levels in the POF model were improved by RGE treatment. Compared with the other three groups, E2 levels in the D-gal group significantly decreased (*p* < 0.05) (Figure 2d) and FSH levels significantly increased (*p* < 0.05) (Figure 2e) on day 42 after administration. In contrast, hormone levels in the D-gal + RGE group were similar to those in the normal control group. The results showed that RGE could ameliorate hormone level changes caused by D-gal in POF mouse models. Importantly, the expression of FSHr protein significantly decreased after D-gal treatment, but was restored to normal levels after RGE treatment (*p* < 0.01) (Figure 2f). These results suggest that RGE alleviated POF symptoms by restoring estrogen levels. In addition, the ovarian indexes of mice in the POF model group were significantly lower (*p* < 0.05). After 28 days of RGE treatment, the ovarian indexes of mice in the RGE groups significantly increased (*p* < 0.05) (Figure 2h,i).

From the results of H&E staining, we found that ovarian follicles and corpus luteum were at different stages of maturation in the normal control group (Figure 2j). In the D-gal group, the ovarian mass decreased and the number of atretic follicles increased significantly—an important characteristic of the effect of POF on follicles. However, after treatment with RGE, the numbers of follicles at different stages of maturation were significantly higher than in the D-gal group.

### 2.3. RGE Inhibited Ovarian Oxidative Stress in Mice with POF

Oxidative stress is associated with the pathological process of aging, which leads to POF. MDA is one of the most important products of lipid peroxidation, indicating the degree of peroxidation. In addition, SOD activity decreases with increasing age and the normal function of the organs changes. Therefore, SOD and MDA levels in the ovarian tissue of mice were evaluated. As shown in the results, SOD activity in the D-gal group decreased significantly and MDA levels were significantly higher than in the normal control group. However, these conditions were reversed after treatment with RGE (*p* < 0.05, *p* < 0.01) (Figure 3a,b). These data suggest that RGE plays an antioxidative role by improving the activity of endogenous antioxidant enzymes.

Next, we verified the antioxidant effect of RGE using western blot analysis. Nuclear factor erythro2-related factor 2 (Nrf2) is a key transcription factor that regulates cellular oxidative stress responses. It acts as a central regulator, maintaining redox balance in the cells and the body. Nrf2 protects cells from damage caused by reactive oxygen species (ROS) and regulates the expression of complex downstream antioxidant enzymes. The expression of Nrf2 and HO-1 proteins significantly decreased after D-gal treatment, but returned to normal after RGE treatment (*p* < 0.05) (Figure 3c). The results showed that RGE could improve D-gal-induced oxidative stress in the female ovary by regulating Nrf2/HO-1.

### 2.4. RGE Mitigated Ovarian Senescence in Mice with POF

To further evaluate the inhibitory effect of RGE on aging-related proteins, we detected the effect of RGE on the expression of p16 protein in ovarian tissues using immunofluorescence staining. The expression level of p16 was low in ovarian tissues of mice in the normal control group, but increased in the model group, and overexpression was significantly reduced in the RGE groups (Figure 4a). Western blot analysis showed that the expression of p53, p21, and p16 proteins in the RGE groups was significantly lower than in the D-gal group (*p* < 0.05) (Figure 4b). These results suggest that RGE can reduce the expression of ovarian senescence-related proteins.

### 2.5. RGE Can Inhibit Ovarian Apoptosis in Mice with POF

Ovarian granulosa cell apoptosis is an important part of POF development. We detected the effect of RGE on Caspase 3 expression in ovarian tissue through immunofluorescence staining. Caspase 3 expression was low in the ovarian tissue of mice in the normal control group but higher in the D-gal group, although this was significantly reduced after RGE treatment (Figure 5a).

Expression of Bax, cl-Caspase 3, and cl-Caspase 9 increased in the D-gal group, while expression of Bcl-2 decreased. These changes were significantly improved after RGE treatment (*p* < 0.05, *p* < 0.01) (Figure 5b). These results suggest that the protective effect of RGE on POF may be achieved through reduction of apoptosis of ovarian cells.

### 2.6. Molecular Docking of RGE and Proteins Associated with Apoptosis Pathways

Relevant studies have shown that after ginseng processing, hydrolysis reactions due to the breaking of glycoside or ester bonds lead to interconversion between different types of saponins, among which red ginseng has a very high content of ginsenoside Rg3, which is one of the representative components in red ginseng with a wide range of pharmacological effects. In this study, to further verify the important role of RGE in the regulation of apoptosis-related proteins, we used ginsenosides Rg3 in water extracts of red ginseng to conduct molecular docking experiments with apoptosis-related proteins with binding energies ≤ 5.0 kJ/mol. To some extent, the ability of Rg3 binding to proteins associated with the apoptosis pathway was verified (Figure 6). The resulting stable complex provides the basis for Bax, Bcl-2, Caspase 3, PI3K, and Akt to act as key proteins in the apoptotic pathway of RGE.

## 3. Discussion

The number of patients with POF has increased due to many factors, including autoimmune abnormalities, radiotherapy and chemotherapy treatments, and increased stress [20]. At present, the pathological mechanism of POF is not clear, thus there is no effective etiological treatment. Currently, HRT is the most common treatment for POF-related symptoms [21].

Gal is a chemical that can be used to stably and efficiently establish animal models of whole-body aging. It is one of the most used methods for constructing animal models of aging due to its advantages of simple drug administration, obvious modeling, high repeatability, and low cost [22,23]. In this study, D-gal successfully induced the POF model, impaired follicular development, decreased the apoptosis rate of granulosa cells, reduced E2 levels, and increased the content of FSH in vivo. Studies have shown that increased galactose content in the blood of females leads to a decrease in the number of ovarian follicles and damages oocytes and ovarian granulosa cells, thus resulting in ovarian failure [24]. Ginseng, a natural Chinese herbal medicine, was obtained after steaming. The types and contents of both saponins and non-saponins [25], which have various pharmacological effects including anti-oxidation and anti-aging [26], increased. In this study, the potential target and mechanism through which RGE improves POF were analyzed using network pharmacology and experimental in vivo animal models.

Combining network topology analysis and PPI network analysis showed that drugs act on multiple targets and signaling pathways. We predicted the core targets and pathways of drug action to be PI3K/Akt, Bax, Bcl-2, and Caspase 3. These pathways and targets were associated with cell proliferation and apoptosis, and these processes affect the occurrence and development of POF [27,28]. Many studies have shown that D-gal can directly induce oxidative stress in vivo, and excessive accumulation of D-gal can lead to galactose toxicity, thus weakening FSH biological activity and inhibiting the production of E2 by granulosa cells [29]. The Nrf2/HO-1 pathway is currently considered to be the most important endogenous antioxidant pathway [30]. ROS are removed by activation of the Nrf2 downstream factor, HO-1, and continuous enzymatic reaction. In addition, the p16 protein is mainly located in ovarian granulosa cells [31] and overexpression of this aging-related protein p16 can induce premature aging of cells, while the anti-aging protein, klotho, inhibits the aging of ovarian and endothelial cells by activating the Nrf2 pathway [32]. In this study, a POF mouse model was induced via intraperitoneal injection of D-gal (400 mg/kg/d) for 42 days and two doses of RGE were given as an intervention on the 15th day. The results showed that RGE could increase SOD activity and decrease MDA content, thus mitigating D-gal-induced oxidative stress. Nrf2 can bind to AREs in the promoter region of Nrf2 target genes, which is an adaptive response to oxidative stress. Excess ROS are removed by activating the Nrf2 downstream factor, HO-1, to produce a continuous enzymatic reaction. The expression of Nrf2 and HO-1 decreased after D-gal treatment, but increased after RGE treatment. Together, these results suggest that RGE successfully alleviates D-gal-induced oxidative damage in the ovary, possibly by eliminating free radicals and stimulating antioxidant enzymes.

Recent studies have reported that oxidative stress plays a crucial role in the development of POF. Sodium arsenite can induce apoptosis of oocytes through oxidative stress and its effect on steroid metabolism [33], which further lead to the atresia of rat follicles. Ovarian granulosa cells, as the basic functional unit in the ovary, are the first to undergo apoptosis in follicles at the later stage of development, inducing oocyte apoptosis and triggering follicular atresia [34]. The PI3K/Akt/mTOR pathway regulates oocyte growth and inhibits ovarian granulosa cell apoptosis [35]. Under oxidative stress, excess ROS can activate the mitochondrial apoptosis pathway resulting in reduced mitochondrial capacity to produce ATP, changes in membrane potential, and increased Ca^2+^ concentration in the cytoplasm, leading to granulosa cell apoptosis [36]. Studies have demonstrated that H_2_O_2_ induced elevation of endogenous pro-apoptosis-related molecules (Bax, Bak) and decreased anti-apoptotic molecules (Bcl-2, Bcl-xL) in ovarian granulosa cells and regulated apoptosis in ovarian granulosa cells through the ROS–JNK–p53 pathway [37,38]. In this study, it was found that after RGE intervention, expression of p-PI3K/p-Akt increased, expression of the pro-apoptotic factors Caspase 3 and Bax decreased, and expression of the anti-apoptotic factor Bcl-2 increased, alleviating the apoptosis of granulosa cells and returning the numbers of follicles in mice ovaries to normal. It was confirmed in vivo that the RGE-activated PI3K/Akt pathway inhibited ovarian granulosa cell apoptosis and increased cell survival rate. RGE also alleviated oxidative stress damage. These suggest that RGE is able to reverse D-gal-induced POF.

Taken together, the results of this study showed that RGE can improve D-gal-induced premature ovarian failure in mice and the mechanism may be associated with the anti-oxidative stress, anti-apoptosis, and anti-aging effect of RGE. This study provides an experimental basis for the application of traditional Chinese medicine to the prevention and treatment of POF as it indicated that RGE can be used as a natural antioxidant to prevent POF.

## 4. Materials and Methods

### 4.1. Network Pharmacology Analysis

#### 4.1.1. Screening Potential Disease and Drug Targets

The main chemical components in ginseng were searched in the TCMSP database (https://tcmspw.com/tcmsp.php/ accessed on 30 January 2023) using the keyword “Renshen”. Oral bioavailability (OB) ≥ 30% and drug-like properties (DL) ≥ 0.18 were used as the qualifying conditions. Active ingredients were obtained after removing ingredients with no clear targets following combination of the results with relevant reports from the literature. To standardize the target-protein information, the target proteins were uniformly transformed into targets in the STRING database (https://cn.string-db.org/ accessed on 30 January 2023).

Target genes associated with premature ovarian failure were searched for in the GeneCards database (https://www.genecards.org/ accessed on 30 January 2023), the OMIM database (http://www.omim.org/ accessed on 30 January 2023), the DrugBank database (https://www.drugbank.com/ accessed on 30 January 2023), and the PharmGkb database (https://www.pharmgkb.org/ accessed on 30 January 2023) using the keyword “premature ovarian failure”.

#### 4.1.2. Protein-Protein Interaction (PPI) Network Construction

To clarify the relationship between the active ingredient targets of red ginseng and the disease targets of POF, Wein diagrams were drawn to identify genes intersecting the active ingredient targets of red ginseng and the targets of POF. The active ingredient targets and the intersecting genes were used to construct an active ingredient–disease target network map using Cytoscape 3.8.0 software.

The intersecting genes were imported into the STRING database (https://string-db.org/ accessed on 30 January 2023) and filtered using a confidence score > 0.9 (high confidence). Unlinked nodes in the network were hidden while the rest of the parameters were kept unchanged. The obtained results were imported into Cytoscape 3.8.0 software in text format to generate protein interaction networks for visualization.

#### 4.1.3. GO and KEGG Enrichment Analyses

GO enrichment analysis and KEGG enrichment analysis were performed using the R cluster Profiler package. GO enrichment analysis analyzed the enrichment levels of proteins and genes from three aspects: biological process (BP), cellular component (CC), and molecular function (MF) [39]. The Kyoto Encyclopedia of Genes and Genomes (KEGG) is a data resource for understanding the high-level functions and utility of biological systems such as cells, organisms, and ecosystems based on molecular-level information, especially large-scale molecular data sets generated through high-throughput experimental techniques such as genome sequencing [40].

#### 4.1.4. Molecular Docking

Related studies have shown that hydrolysis reactions occurring due to breakage of glycosidic or ester bonds after ginseng processing lead to interconversion of different saponin types. Rg3 is one of the most representative components of red ginseng [41] and has a wide range of pharmacological effects. We verified the ability of 20(S)-ginsenoside Rg3 binding to key targets in POF using molecular docking experiments. The above compound was identified in the PubChem database (https://pubchem.ncbi.nlm.nih.gov/ accessed on 30 January 2023) and the crystal structures of Bax, Bcl-2, Caspase 3, PI3K, and Akt were obtained—in PDB format—from the Protein Data Bank database (http://www.rcsb.org/ accessed on 30 January 2023). Protein and ligand data were saved in PDBQT format using AutoDockTools-1.5.6. Finally, molecular docking was performed using AutoDockVina.exe and the results were calculated. PyMOL software was used to show docking results of key target proteins with the strongest abilities to bind the compounds.

### 4.2. Preparation of RGE

After purification, fresh ginseng was autoclaved at 120 °C and 0.1 kPa for 30 min, then naturally cooled to room temperature and put into a blast drying oven set at 60 °C and dried for 24 h to obtain red ginseng. The prepared red ginseng samples were crushed and passed through 80 mesh sieves to obtain powders. Distilled water was added to achieve a material–liquid ratio of 1:20 and the extract was extracted using three rounds of ultrasonication (60 min each). The temperature was controlled at about 45 °C. The extracts were combined and filtered and then concentrated under reduced pressure in a BK-RE-1A rotary evaporator. Samples were obtained after concentrating and freeze-drying.

### 4.3. Design of Animal Experiments

7-week-old female ICR mice (weighing 24–26 g) were bought from Changchun YISI Biotechnology Co., Ltd. (Certificate No. of SCXK (JI)-2020-0002). The controlled living environment of the mice was maintained at a temperature of 22 ± 2 °C, relative humidity of 60 ± 10%, and a 12 h light/dark cycle. The mice were given unrestricted access to food and drinking water. All animal handling and experimental procedures were in accordance with the approval of the Experimental Animal Ethics Committee of Jilin Agricultural University.

All female ICR mice were randomly divided into 4 groups, with 15 mice in each group: normal control group, D-gal group, D-gal + RGE low-dose group (200 mg/kg) (Beijing Solaibao Technology Co., Ltd., Beijing, China), and D-gal + RGE high-dose group (400 mg/kg). The D-gal group was intraperitoneally injected with D-gal (400 mg/kg/d) for 42 consecutive days. The D-gal + RGE low-dose group and the D-gal + RGE high-dose group were injected with D-gal subcutaneously (400 mg/kg) for 42 consecutive days, and intraperitoneally injected with the corresponding dose of RGE daily for 28 consecutive days from the 15th day of the first D-gal injection. Vaginal secretions were collected from the mice from 8:00 to 9:00 a.m. every day and changes in the estrous cycle were observed using H&E staining.

### 4.4. Ovarian Index Measurement

After blood collection, the bilateral ovaries of mice were taken and the ovary index (ovarian wet weight/body weight) was determined. After stripping the surrounding adipose tissue, the ovaries were placed in frozen tubes and stored at −80 °C in a refrigerator for subsequent experiments.

### 4.5. Determination of Biochemical Parameters

After dissecting the mice, ovarian samples were collected and lysed on ice for 120 min, centrifuged at 12, 000 rpm for 10 min, and the supernatant collected. SOD activity and MDA levels in the supernatant were quantified using commercially available kits (Nanjing Jiancheng Bioengineering Institute, Nanjing, China) according to the manufacturers’ protocols. Absorbance was measured at the corresponding wavelength using an automatic microplate reader (Bio Tek Elx800, Berton Instruments, VT, USA). To detect sex hormones such as E2 and FSH (Uscn Life Science Inc., Wuhan, China), blood samples were obtained from anesthetized mice through a retroorbital puncture on the last day of administration and serum was collected after centrifugation at 3500 rpm for 15 min. Estrogen levels were determined using the ELISA kit according to the manufacturer’s instructions.

### 4.6. Ovarian Histomorphology

The ovaries were fixed in 4% formaldehyde for 24 h, then dehydrated, paraffin-embedded, and cut into 5-μm sections. The sections were stained with hematoxylin and eosin (H&E) after rehydration and morphological observation and classification of follicles were conducted.

### 4.7. Immunofluorescence

The ovarian tissue sections were dewaxed and washed in distilled water, then dried and incubated with primary antibodies against p16 (1:500) and Caspase 3 (1:500) overnight at 4 °C. The next day, once the samples returned to room temperature, the primary antibody was rinsed off with pre-configured PBS, Alexa Fluor 594 (Thermofisher, Shanghai, China) was added, and the samples incubated in the dark for 30 min. The ovarian nucleus was then stained with DAPI. Finally, anti-fluorescence quenching agent (PVB) was used to seal the plates, expressions of the aging marker, protein p16, and the apoptosis protein, Caspase 3, were observed under a fluorescence microscope (Leica DM2500, Wetzlar, Germany), and fluorescence density was analyzed.

### 4.8. Western Blot Analysis

The fat surrounding the mouse ovarian tissue was shaved off and the tissue samples fully ground and placed in a pre-prepared mixed RIPA lysate (Mengbio, Chongqing, China). Total protein concentration in ovarian tissue was determined using the BCA protein assay (Beyotime Biotechnology, Shanghai, China) and quantified to the same concentration. Each lane was sampled with 13 µg equivalent proteins on 10 or 15 SDS-PAGE and transferred to the PVDF membrane (0.22 µm, 0.45 µm). The PVDF membrane was blocked for 2 h with 5% skim milk prepared with TBS-T (TBS + 0.1% Tween-20) and then washed 3 times (8 min each time) with TBS-T lotion. Subsequently, the membranes were incubated overnight at 4 °C with primary antibodies against PI3K (1:500), p-PI3K (1:500), Akt (1:500), p-Akt (1:500), FSHr (1:1000), p53 (1:1000), p21 (1:1000), p16 (1:1000), klotho (1:1000), Nrf2 (1:1000), HO-1 (1:1000), Keap1 (1:1000), Bax (1:1000), Bcl-2 (1:1000), Caspase 9 (1:1000), cl-Caspase 9 (1:1000), Caspase 3 (1:1000), and cl-Caspase 3 (1:1000). The levels of GAPDH (1:1000) or β-actin (1:1000) were assessed as loading controls. The antibodies against PI3K, p-PI3K, Akt, p-Akt, p53, Bax, Caspase3, cl-Caspase 3, GAPDH, and β-actin were purchased from Cell Signaling Technology while the antibodies against p21, p16, Nrf2, HO-1, Keap1, Bcl-2, Caspase 9, and cl-Caspase 9 were purchased from Proteintech. In addition, antibodies against FSHr and klotho were purchased from Wanleibio Co., Ltd., Shenyang, China. The next day, the strips were cleaned with TBST and the secondary antibody (1:2000) and the primary antibody were coupled at room temperature for 2 h. The signal was captured by an emitter-coupled logic (ECL) substrate (Pierce Chemical Co., Rockford, IL, USA). Image J 6.0 software (NIH, Bethesda, MD, USA) was used to quantitatively analyze the intensity of each band.

### 4.9. Statistical Analysis

All data were recorded as mean ± standard deviation (Mean ± SD) and studied using one-way analysis of variance (ANOVA) and Bonferroni post hoc test. Graphs of the statistical data were generated using GraphPad Prism 8.0.2 software (GraphPad Software, Inc., San Diego, CA, USA). In all cases, *p* < 0.001, *p* < 0.01, and *p* < 0.05 were considered to be significant.

## Figures and Tables

**Figure 1 ijms-24-08210-f001:**
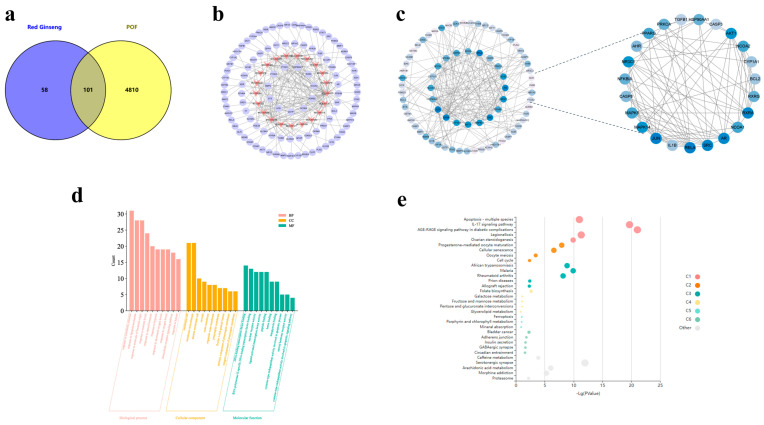
Visualization of interacting targets in the active components of red ginseng and POF. Venn diagram of the numbers of overlapping and specific targets in red ginseng pairs (blue circle) and POF (yellow circle) (**a**); complex target pathway network of red ginseng against POF. The pink V-shaped node is the active component in red ginseng and the blue node is the potential target (**b**). The selected core objectives (**c**); GO enrichment of red ginseng against POF and KEGG pathway analysis of POF targets (**d**,**e**).

**Figure 2 ijms-24-08210-f002:**
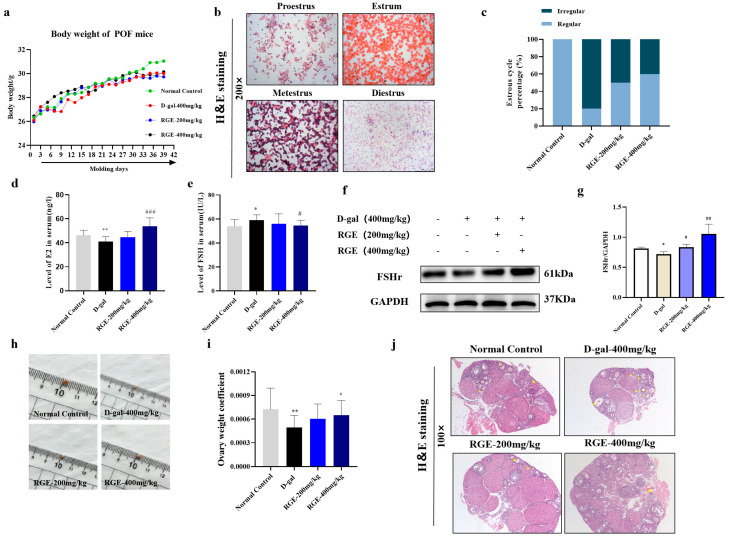
Effect of RGE on the estrous cycle and estrogen and ovarian histopathology. Effect of RGE on body weights of mice with POF (**a**); Effect of RGE on estrous cycle of mice with POF (**b**,**c**); The effect of RGE on the ovarian indexes of mice with POF(**h**,**i**); Effects of RGE on serum E2 (**d**) and FSH levels in mice (**e**); Effects of RGE on estrogen receptor expression (**f**,**g**); Histological examination of the ovary using H&E staining (100×) (**j**); yellow triangles represent atretic follicles, D-gal group, low-dose RGE, and high-dose RGE. Data are expressed as Mean ± SD (*n* = 15). * *p* < 0.05, ** *p* < 0.01 vs. normal group; ^#^ *p* < 0.05, ^##^ *p* < 0.01, ^###^ *p* < 0.001 vs. D-gal group.

**Figure 3 ijms-24-08210-f003:**
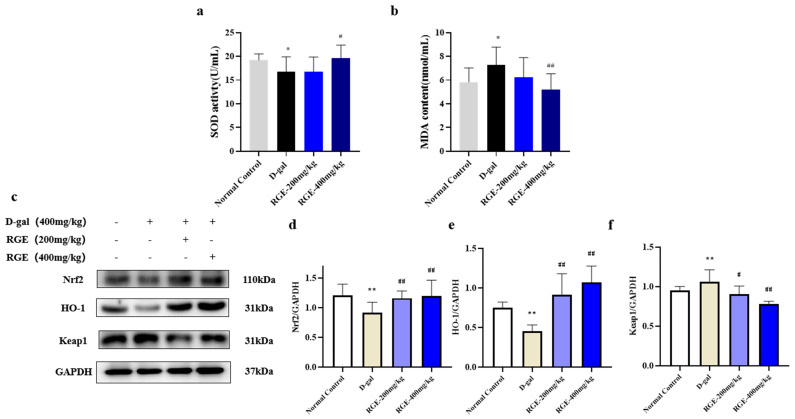
Effect of RGE on oxidative stress. SOD levels (**a**); MDA levels (**b**); Western blot analysis of the Nrf2/HO-1 signaling pathway (**c**–**f**). Data are expressed as Mean ± SD (*n* = 15). * *p* < 0.05, ** *p* < 0.01 vs. normal group; ^#^ *p* < 0.05, ^##^ *p* < 0.01 vs. D-gal group.

**Figure 4 ijms-24-08210-f004:**
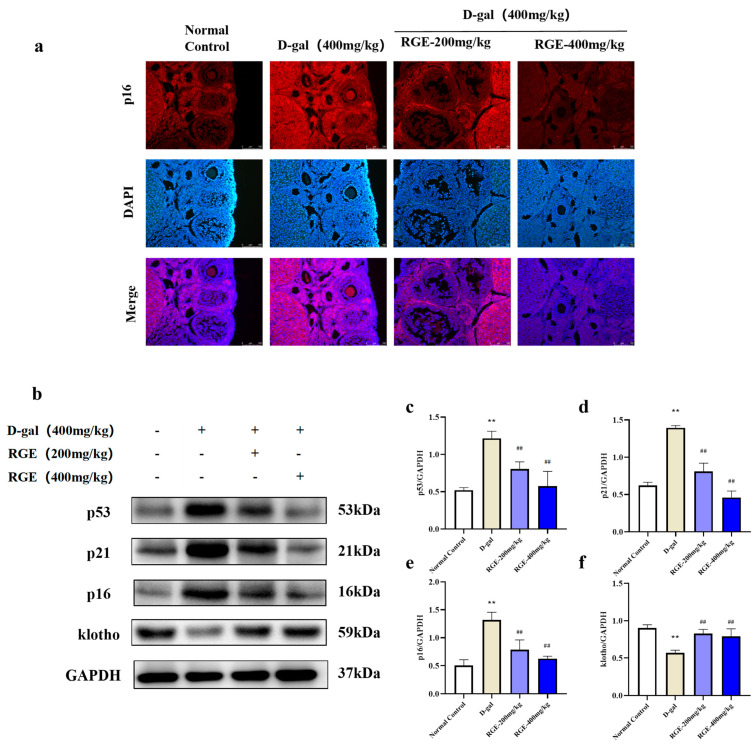
Effect of RGE on ovarian senescence in mice with POF. p16 immunofluorescence staining (**a**); Expression of p53, p21, p16, and klotho proteins was analyzed by western blotting with specific primary antibodies (**b**–**f**). Data are expressed as Mean ± SD (*n* = 15). ** *p* < 0.01 vs. normal group; ^##^ *p* < 0.01 vs. D-gal group.

**Figure 5 ijms-24-08210-f005:**
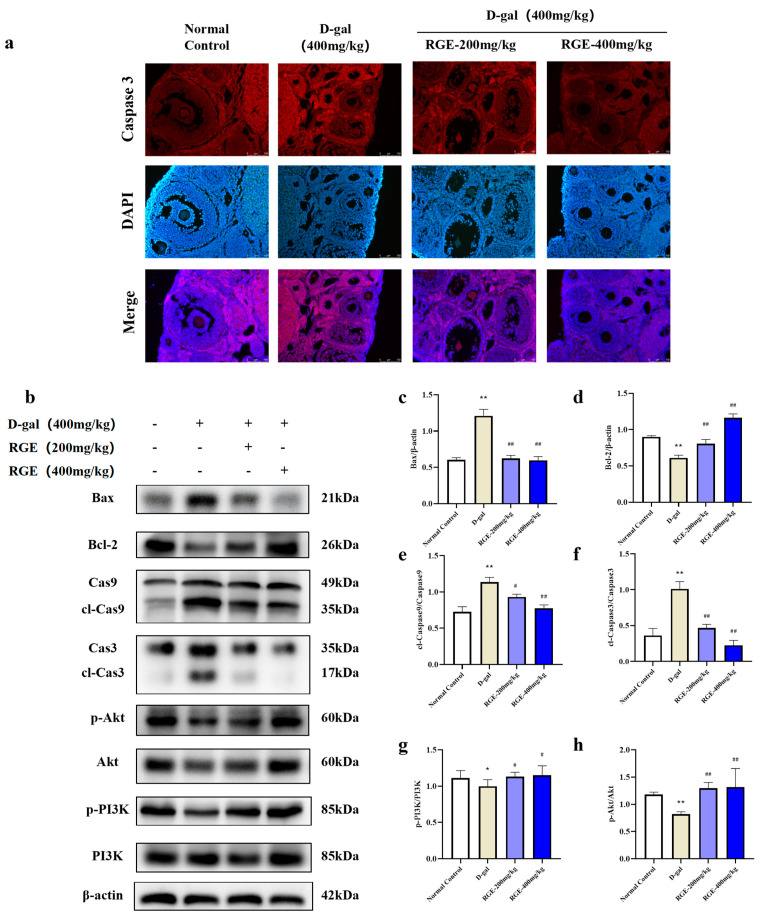
Inhibitory effect of RGE on ovarian apoptosis in mice with POF. Caspase 3 immunofluorescence staining (**a**); Expression of Bax, Bcl-2, Caspase 3, cleaved-Caspase 3, Caspase 9, cleaved-Caspase 9, PI3K, p-PI3K, Akt, and p-Akt proteins was analyzed by western blotting with specific primary antibodies (**b**–**h**). Data are expressed as Mean ± SD (*n* = 15). * *p* < 0.05, ** *p* < 0.01 vs. normal group; ^#^ *p* < 0.05, ^##^ *p* < 0.01 vs. D-gal group.

**Figure 6 ijms-24-08210-f006:**
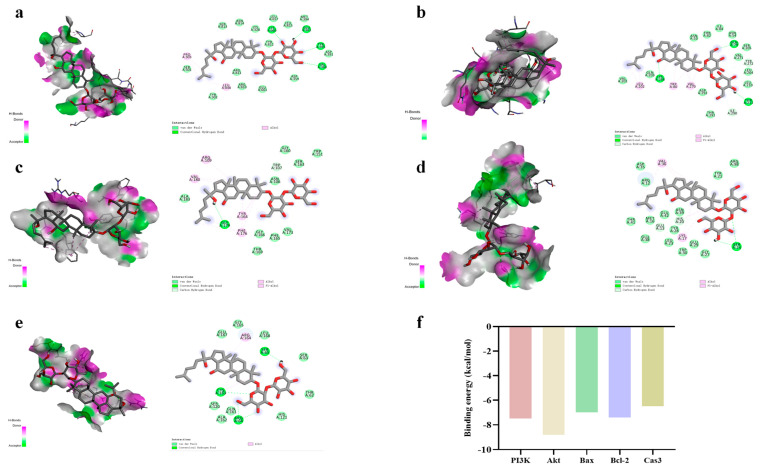
Molecular docking interactions between 5 core targets and RGE. Binding modes of RGE to PI3K (**a**), Akt (**b**), Bax (**c**), Bcl-2 (**d**), and Caspase 3 (**e**). Compound protein binding ability scores (**f**).

**Table 1 ijms-24-08210-t001:** Active Ingredients in RG.

	Herb	ID	Ingredients	OB%	DL
1	Ginseng	MOL002879	Diop	43.59	0.39
2	Ginseng	MOL000449	Stigmasterol	43.83	0.76
3	Ginseng	MOL000358	beta-sitosterol	36.91	0.75
4	Ginseng	MOL003648	Inermin	65.83	0.54
5	Ginseng	MOL000422	Kaempferol	41.88	0.24
6	Ginseng	MOL004492	Chrysanthemaxanthin	38.72	0.58
7	Ginseng	MOL005308	Aposiopolamine	66.65	0.22
8	Ginseng	MOL005314	Celabenzine	101.88	0.49
9	Ginseng	MOL005317	Deoxyharringtonine	39.27	0.81
10	Ginseng	MOL005318	Dianthramine	40.45	0.20
11	Ginseng	MOL005320	Arachidonate	45.57	0.20
12	Ginseng	MOL005321	Frutinone A	65.90	0.34
13	Ginseng	MOL005344	Ginsenoside Rh2	36.32	0.56
14	Ginseng	MOL005348	Ginsenoside Rh4	31.11	0.78
15	Ginseng	MOL005356	Girinimbin	61.22	0.31
16	Ginseng	MOL005357	Gomisin B	31.99	0.83
17	Ginseng	MOL005360	Malkangunin	57.71	0.63
18	Ginseng	MOL005376	Panaxadiol	33.09	0.79
19	Ginseng	MOL005384	Suchilactone	57.52	0.56
20	Ginseng	MOL005399	Alexandrin_qt	36.91	0.75
21	Ginseng	MOL005401	Ginsenoside Rg5	39.56	0.79
22	Ginseng	MOL000787	Fumarine	59.26	0.83
23	Ginseng	MOL005331	Ginsenoside Rb1	6.24	0.04
24	Ginseng	MOL005333	Ginsenoside Rb2	6.02	0.04
25	Ginseng	MOL005343	Ginsenoside Rg3	29.69	0.77
26	Ginseng	MOL005344	Ginsenoside Rh2	36.32	0.56
27	Ginseng	MOL005338	Ginsenoside Re	4.27	0.12

## Data Availability

The data that support the findings of this study are available from the corresponding author upon reasonable request.

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
