# Peer review of "Red Ginseng Improves D-galactose-Induced Premature Ovarian Failure in Mice Based on Network Pharmacology"

_ijms, 2023, doi:10.3390/ijms24098210_

Round 1
Reviewer 1 Report
The authors have provided us a very interesiting manuscript titled "Red Ginseng Improves D-galactose Induced Premature Ovarian Failure in Mice Based on Network Pharmacology" . However, I would like to request some revisions to increase the transparency and reproducibility of your research.
Firstly, I suggest that the authors provide more detailed information on how the activity of SOD and content of MDA were determined. It would be helpful if you include a sentence in your Methods section that specifies the method used for the chemical colorimetry. Additionally there are several reagents, and equipment whose manufacturer are not identified. This information is important for readers who want to replicate your results and methods.
Secondly, I recommend that you acknowledge the limitations of your study. By discussing the potential limitations of your methodology, or data analysis techniques, you can help readers to understand the scope and generalizability of your findings.
Thank you for considering my suggestions.
Sincerely,
Language is fine. Please provide reagents and equipment manufacturer identification.
Reviewer 2 Report
Sound laboratory work. However, I would have liked to see more "selling" of this work in the abstract and introduction. I think after this is "tweaked" the overall manuscript quality will improve.
What is the main question addressed by the research? Can Red Ginseng decrease (treat) premature ovarian failure, based on prediction results from network pharmacology.
Is it relevant and interesting? Yes, the paper is relevant. However, I believe the authors focused too much on describing the technical work, e.g., experience, and not why they were doing it. I think they could explore more background with network pharmacology.
How original is the topic? The topic is somewhat original. People have been studying natural products for decades. However, the use of network pharmacology does stand out in this paper.
What does it add to the subject area compared with other published material? I believe it does add. However, I don’t think it’s ground breaking.
Is the paper well written? Yes, the paper is well written. There may be grammatical errors that I could have overlooked. I would suggest a second look by the authors.
Is the text clear and easy to read? Yes, I see no issues with the text or the way the manuscript is arranged.
Are the conclusions consistent with the evidence and arguments presented? Yes, again the paper Is presented technically sound.
Do they address the main question posed? Yes, they do. As far as experiments and validations of those experiments. However, as my previous submitted review, I think they can do a better job at selling the research. I think an improvement in the abstract and introduction would Mae this paper more interesting.
I believe the English is fine. May want to review for minor errors, e.g., grammar and punctuation.
